# Long-Term Care Admissions Following Hospitalization: The Role of Social Vulnerability

**DOI:** 10.3390/healthcare7030091

**Published:** 2019-07-15

**Authors:** Judith Godin, Olga Theou, Karen Black, Shelly A. McNeil, Melissa K. Andrew

**Affiliations:** 1Division of Geriatric Medicine, Nova Scotia Health Authority, Dalhousie University, 5955 Veterans’ Memorial Lane, Halifax, NS B3H 2E1, Canada; 2Department of Community Health and Epidemiology, Dalhousie University, Halifax, NS B3H 4R2, Canada; 3Division of Infectious Diseases, Dalhousie University, Halifax, NS B3H 4R2, Canada

**Keywords:** social vulnerability, frailty, long-term care placement, social vulnerability index, frailty index

## Abstract

We sought to understand the association between social vulnerability and the odds of long-term care (LTC) placement within 30 days of discharge following admission to an acute care facility and whether this association varied based on age, sex, or pre-admission frailty. Patients admitted to hospital with acute respiratory illness were enrolled in the Canadian Immunization Research Network’s Serious Outcomes Surveillance Network during the 2011/2012 influenza season. Participants (N = 475) were 65 years or older (mean = 78.6, SD = 7.9) and over half were women (58.9%). Incident LTC placement was rare (N = 15); therefore, we used penalized likelihood logistic regression analysis. Social vulnerability and frailty indices were built using a deficit accumulation approach. Social vulnerability interacted with frailty and age, but not sex. At age 70, higher social vulnerability was associated with lower odds of LTC placement at high levels of frailty (frailty index (FI) = 0.35; odds ratio (OR) = 0.32, 95% confidence interval (CI) = 0.09–0.94), but not at lower levels of frailty. At age 90, higher social vulnerability was associated with greater odds of LTC placement at lower levels of frailty (FI = 0.05; OR = 14.64, 95%CI = 1.55, 127.21 and FI = 0.15; OR = 7.26, 95%CI = 1.06, 41.84), but not at higher levels of frailty. Various sensitivity analyses yielded similar results. Although younger, frailer participants may need LTC, they may not have anyone advocating for them. In older, healthier patients, social vulnerability was associated with increased odds of LTC placement, but there was no difference among those who were frailer, suggesting that at a certain age and frailty level, LTC placement is difficult to avoid even within supportive social situations.

## 1. Introduction

When people are admitted to hospital, the hope is that they will recover and be able to function at a level comparable to how they functioned before they became ill. As people age and become frail, their ability to recover from illness becomes hampered [1], and hospitalization of older frail adults increases the odds of being admitted to a long-term care (LTC; also known as a nursing home or care home) facility [2]. Most people want to age at home rather than in an LTC facility [3]. Frail people with a lot of social resources may be able to forestall a move to an LTC facility longer than individuals who have fewer social and economic resources.

Some social factors are associated with a higher risk of institutionalization even after controlling for more traditional predictors such as cognition, age, and activities of daily living [4,5,6,7]; however, only a limited range of social factors has been examined, and these social factors are typically considered as individual variables [8,9]. A recent systematic review and meta-analysis of predictors of LTC placement following acute hospitalizations found that social support, availability and costs of social care, and loneliness were rarely considered [10]. The costs associated with care can be an important consideration for patients and family members making decisions about whether and what type of care they seek [11]. As such, financial security could be a deciding factor in entering long-term care. In Canada, where subsidies for LTC are sometimes available but patients are often expected to pay some portion of the costs according to their means, an understanding of an individual’s full social and economic circumstances is needed. Much more research is needed to understand how a person’s complete constellation of social factors may affect their risk of LTC placement. 

Examining social factors one at a time can be advantageous, as this type of knowledge may more easily identify targets for interventions. On the other hand, examining these factors in isolation does preclude an understanding of how social factors combine to influence outcomes such as LTC placement. For instance, living alone for someone with a strong sense of community and an accessible external social support network may incur a different level of risk compared to someone living alone and who is also isolated within their community. Social vulnerability is the extent to which overall social circumstances leave individuals susceptible to negative health outcomes [12,13]. 

Given that past research has only examined social factors independently, we sought to understand the association between social vulnerability measured comprehensively in a single index and the odds of incident placement in a LTC facility within 30 days of discharge following admission to an acute care facility. Further, because it is unknown whether the effects of social factors on LTC placement are consistent across sex, age, and frailty, we tested whether frailty, age, or sex moderated the association between social vulnerability and odds of LTC placement.

## 2. Methods

### 2.1. Participants

Patients admitted to hospital with acute respiratory illness were enrolled in the Canadian Immunization Research Network’s Serious Outcomes Surveillance Network during the 2011/2012 influenza season. During this season, the study included surveillance of 16,000 beds in 38 hospitals across 6 Canadian provinces. We conducted a secondary analysis of this dataset for the current study. The availability of reliable data on LTC both prior to admission, at discharge, and 30 days post discharge in addition to the social and health variables made this an ideal dataset to answer our research question. We included patients who were aged 65 years and older and who had a complete frailty index (FI) and social vulnerability index (SVI). Our main analyses included participants (N = 475) who were not in LTC prior to admission and who were either discharged home or placed in LTC at discharge or within 30 days. We also ran three sensitivity analyses (1) including participants who were in LTC prior to admission (N = 501); (2) including all participants regardless of discharge disposition (e.g., transferred to other units/hospitals or died) but excluding those who were in LTC prior to admission (N = 515); and (3) including all participants regardless of discharge disposition (e.g., transferred to other units/hospitals or died) and those who were in LTC prior to admission (N = 542).

### 2.2. Measures

The age and sex of participants were recorded at admission. Both frailty and social vulnerability were measured using a deficit accumulation approach such that appropriate variables for each index were coded 1 for deficit present or 0 for deficit absent. Order response categories were assigned intermediary values (e.g., 0.5). For each index, the number of deficits present for each individual was summed and divided by the number of deficits considered. Indices were created for every individual who had data for at least 80% of the deficits. 

We created an 18-deficit social vulnerability index (SVI) for use in the clinical setting of the SOS Network that included deficits reflecting living situation, social support, socioeconomic status, social engagement, and perceived control over one’s life (Table 1). The SVI was designed for use in the current study based on previous research validating the SVI in epidemiological and population settings [12,14].

Frailty was measured using a 39-deficit frailty index (FI) that was previously validated in this dataset [15]. The first frailty assessment reflected the participants’ state two-weeks prior to admission (baseline) and the second assessment reflected their state at admission. The FI was created based on the guidelines outlined by Searle and colleagues and included items covering cognition and dementia, functioning, mood, sensory deficits, and illness [16]. 

To assess whether participants were placed in LTC within 30 days of discharge, we considered information about discharge from acute care and residence at 30 days post discharge. At the time of hospital discharge, the place to which participants were discharged was recorded according to the following nine categories: transferred for completion of treatment, management of sequelae, palliative care, or rehabilitation, recovered but remained in hospital for other reason, discharged home, alternative level of care, died during admission, or transferred to LTC. Participants were contacted 30 days post-discharge for updated information, and place of residence was categorized according to the following five categories: LTC facility, private house, apartment, rooming house, shelter, or other. Participants who were placed in LTC at discharge or were in an LTC facility at 30 post-discharge were categorized as being placed in LTC. Participants were considered as discharged home if they returned home at discharge and were not in LTC 30 days post discharge. 

### 2.3. Statistical Analyses

Descriptive statistics were used to describe the sample: means and standard deviations for continuous variables and frequencies and percentages for categorical variables. Due to the rarity of incident LTC placement in the current sample (N = 15 in the main analysis), we used Firth’s penalized likelihood logistic regression to examine the association between social vulnerability and odds of LTC placement, which removes small sample bias [17,18]. Analyses were conducted using in R [19] using the logistf package [20].

The initial logistic regression included all variables (i.e., age, sex, social vulnerability, baseline frailty, and admission frailty) and three interaction terms: sex by social vulnerability, frailty by social vulnerability, and age by social vulnerability. The FI and SVI were multiplied by 10 prior to inclusion in the regression so that coefficients and odds ratios could be interpreted in terms of a 0.1 change in the indices [21,22]. 

For the statistically significant interactions (i.e., frailty and age by social vulnerability), we explored the simple associations of social vulnerability at four levels of frailty (i.e., 0.05, 0.15, 0.25, 0.35) and at three different ages (i.e., 70, 80, and 90 years). We followed the procedures outlined by Aiken and West to explore the simple associations [23]: for example, to estimate the association of social vulnerability at age 70, a new age variable was created (age—70) and the interaction was recalculated using this new age variable. The logistic regression was rerun and the coefficient and odds ratio for social vulnerability in this follow-up regression represents the association between social vulnerability and odds of LTC placement at age 70. We repeated this procedure for each combination of the four levels of frailty and three different ages.

## 3. Results

The mean age of participants was 78.6 (SD = 7.9) and over half were women (58.9%). Just over 5% of the sample had dementia. Descriptive statistics for the participants included in the main and sensitivity analyses are included in Table 2.

Controlling for age, sex, and baseline and admission frailty, the association of social vulnerability with odds of LTC placement was significantly moderated by age and baseline frailty, but not sex. Social vulnerability was associated with increased odds of LTC placement in more fit (b = −0.70, *p* < 0.01) and older (b = 0.09, *p* < 0.05) individuals. For every 0.1 increase in the FI, the unstandardized coefficient for the SVI decreased by 0.7. For every 1-year increase in age, the unstandardized coefficient for the SVI increased by 0.09. See Table 3 for the logistic regression results for the main and sensitivity analyses. 

At age 70, social vulnerability was associated with lower odds of LTC admission at high levels of frailty (FI = 0.35; OR = 0.32, 95%CI = 0.09–0.94), but not at lower levels of frailty. At age 90, social vulnerability was associated with greater odds of LTC admission in the fittest people (FI = 0.05; OR = 14.64, 95%CI = 1.55, 121.21 and FI = 0.15; OR = 7.26, 95%CI = 1.06, 41.84), but not at higher levels of frailty. See Table 4 for all simple associations from the main analyses.

## 4. Discussion

In this paper, we considered social factors holistically using a single comprehensive measure. We found that the association between social vulnerability measured holistically and LTC placement was moderated by age and frailty, but not sex. Specifically, at younger ages (e.g., 70 years), social vulnerability was associated with lower odds of LTC placement for those who were the frailest, while at older ages (e.g., 90 years), social vulnerability was associated with increased odds of LTC placement in those adults who were non-frail or only mildly frail but did not impact odds of LTC placement among the frailest participants.

Research examining the associations between social factors and LTC placement has typically examined a single social factor at a time or a limited number of social factors treated as individual variables. For instance, one study found that a care receiver’s sense of social cohesion with their community was associated with higher odds of remaining in the community over five years even after taking into consideration dementia and physical functioning [4]. In a Canadian study, clients of assisted living facilities were at higher risk of LTC placement if they had poor social relationships and had only limited engagement with activities, on top of the increased risk due to mild or severe cognitive impairment, activities of daily living impairment, health problems, falls, hospitalizations, and incontinence [5]. In an Australian study of older men, men with high social interactions and those who were married were less likely to be institutionalized within 3.4 years, controlling for cognition, activities of daily living, and grip strength, and age [6]. None of these studies tested whether these relationships between social factors and LTC placement varied based age, sex, or degree of frailty.

Although examining social factors one at a time provides useful information and can identify specific targets for intervention, there are benefits to using a comprehensive instrument. Any individual social factor may not be associated with LTC placement when considered on its own but may be an important predictor in the presence of other social deficits [13]. A study using data from the English Longitudinal Study on Ageing (ELSA) found that two different measures of loneliness predicted moving into a care home even after controlling for cognition, disability, physical health, wealth, age, sex, depression, and social isolation [7]. Interestingly, social isolation, one of the covariates in the ELSA study on loneliness, was not a significant predictor of moving into a care home in the fully adjusted model. Possibly, the effects of loneliness may be stronger at higher levels of social isolation or social isolation may be associated a move to a LTC facility only at when combined with other social deficits. 

In one study, researchers examined whether education, income, residential area and social support could be used to create social profiles that could predict LTC usage [24]. They found support for two latent classes: (1) more education, satisfied with household income, and urban residence versus (2) less education, less satisfaction with income, and rural residence. Although these latent classes represent a more comprehensive view of social factors than is typically considered, these latent classes were not significantly associated with LTC usage. Possibly, a dichotomous variable such as the one created in the above study did not capture enough variability in the extent that people are socially vulnerable. Using a deficit accumulation approach, such as the SVI, offers the advantage of providing a more finely tuned, comprehensive and graded measure of social vulnerability. To our knowledge, this is the first use of the SVI approach in a clinical setting, although the SVI approach has been adapted and validated in preparation for clinical use in other settings [25].

In our study, we found that the association between social vulnerability and LTC placement was different depending on frailty and age. Our results suggested that in older adults (e.g., 90 years), social factors only play a role in LTC placement for individuals with fewer health problems, suggesting that perhaps once there are a critical number of health issues, social factors no longer play a role. At the younger end of the spectrum (e.g., 70 years), being socially vulnerable was associated with lower odds of LTC placement in those participants who were the frailest, and hence, presumably most in need of LTC. Also, a jurisdiction’s policies for LTC payment and subsidies could influence decision-making; for younger, socially vulnerable patients who may be in need of LTC, perceptions of a lengthy and costly stay may deter the patient and their family members from considering LTC. Interactions between health and social factors on LTC placement are infrequently examined; however, one study found that osteoporosis interacted with social factors such that osteoporosis was only associated with LTC usage in a group of participants classed together by low education, dissatisfaction with income, rural residence, and social support but not in the other class [24]. There were no other significant interactions between social factors and the other chronic diseases considered by the authors. 

Although not a direct test of moderation, researchers sometimes conduct separate analyses for men and women, suggesting potential moderation. For example, a Canadian study which linked data from the Canadian Community Health Survey to the 2011 Canadian Census found that living alone was associated with higher odds of living in a nursing home for women, but not men [26]. They also found that, for both women and men, the odds of living in a nursing home were higher for individuals who had lost their spouse through death or separation, were not married, did not own their home, had spent time in hospital or convalescent home, or were diagnosed with dementia. In our study, the interaction between sex and social vulnerability was not statistically significant; however, we may have lacked the necessary statistical power to detect an interaction due to the rarity of LTC placement in our sample. Future researchers should continue to examine sex and gender differences in the association between social factors and LTC placement.

The phrase “right care, right time, right place” has been used as a guide for quality health care [27], and it is particularly pertinent in the context of LTC. Ideally, individuals can access LTC when they need it and can age in place when possible and preferred. Our results suggest that this may not always be the case: younger, frail, socially vulnerable patients may not have access to LTC when they need it and older, less frail patients who are socially vulnerable may be placed in LTC prematurely. This is important information for policy makers who are trying to optimize access to LTC and shorten waitlists. 

Our results should be interpreted with caution. Due to the small number of participants who were placed in LTC, our estimates of the odds of LTC placement are imprecise—as evidenced by the wide confidence intervals. We ran a number of sensitivity analyses to help address this limitation, and the results were not substantially different in any of these analyses. So, while we have some evidence supporting the role of social vulnerability in LTC placement, further research is needed to understand the strength of association between social vulnerability and LTC placement following hospitalization. Data for the social vulnerability index was only available for the 2011–2012 influenza season. As such, we are not able to compare between seasons, and we do not know the extent to which our results may have been influenced by period effects including the particular influenza strains that were circulating during this season. Further, there were a number of factors that we were unable to consider in our study, such as caregiver health and caregiver burden, that are likely associated with LTC placement. We considered frailty in our study as a moderator of the relationship between social vulnerability and odds of LTC placement; however, we did not examine what factors could protect against LTC placement in frailer adults. This is an important avenue for future research. The relationship between LTC and acute care varies by jurisdiction within Canada, which we are not able to account for specifically. Additionally, there have likely been some changes in health care policy in the time since our data were collected, although it is unlikely that this would have systematically affected our results. Some well publicized recent changes in Canadian health care policy include the legalization of cannabis and Medical Assistance in Dying. Further research into the association between social factors and LTC placement as social and health policies evolve will continue to be important. Even with these limitations in mind, our study contributes to the literature into the link between social factors and LTC placement by highlighting that the social context interacts with health in complex ways in predicting LTC placement.

## 5. Conclusions

Through the current study, we learned that the association between social vulnerability and odds of LTC placement is complex and likely differs between groups of people. The association between social vulnerability and LTC placement was moderated by frailty and age but not by sex. Social vulnerability was associated with lower odds of LTC placement in younger, frail patients. This may be because, although these frailer participants might be in need of LTC, they may not have anyone advocating for them. In contrast, social vulnerability was associated with greater odds of LTC placement in older patients who were fitter, suggesting that reducing social vulnerability through the provision of social resources and policy may help older fit adults age in place for longer and avoid or delay LTC placement. 

## Figures and Tables

**Table 1 healthcare-07-00091-t001:** Social vulnerability index (SVI) items and deficit coding.

Item	Coding
Lives alone	0 = no; 1 = yes
Current marital status	0 = married or common law; 1 = single, divorced or widowed
Highest level of education	0 = post-secondary (college, university bachelor, graduate or professional degree; 0.33 = trades or apprenticeship; 0.67 = high school; 1 = less than high school
Ever homeless	0 = no; 1 = yes
Lives in a rooming house, group home, shelter, or is currently homeless	0 = no; 1 = yes
Feels that income currently satisfies needs	0 = yes; 1 = no
How often does the patient participated in activities, groups, or clubs in the community	0 = often (weekly); 0.5 = sometimes; 1 = never
Does the patient volunteer in the community	0 = yes; 1 = no
How often does the patient attend religious services	0 = often (weekly); 0.5 = sometimes; 1 = never
Does the patient have someone to count on for help or support	0 = yes; 1 = no
Does the patient feel they need more help or support	0 = no; 1 = yes
Does the patient have someone to confide in	0 = yes; 1 = no
How often does the patient get together and socialize with family/relatives	0 = often (weekly); 0.5 = sometimes; 1 = never
How often does the patient get together and socialize with friends	0 = often (weekly); 0.5 = sometimes; 1 = never
Does the patient feel lonely	0 = no; 1 = yes
Does the patient say that most people can be trusted	0 = yes; 1 = no
Does the patient feel safe their neighborhood	0 = yes; 1 = no
Does the patient feel they have control over things that happen to them	0 = yes; 1 = no

**Table 2 healthcare-07-00091-t002:** Means (standard deviations) for continuous variables and N (%) for dichotomous variables for the main and sensitivity analyses.

	Main Analyses N = 475	Sensitivity Analysis 1 N = 501	Sensitivity Analysis 2 N = 515	Sensitivity Analysis 3 N = 542
Women (%)	280 (58.9)	297 (59.3)	300 (58.3)	317 (58.5)
Age (SD)	78.6 (7.9)	79.0 (8.0)	78.7 (8.0)	79.1 (8.1)
LTC placement prior to admission (%)	NA	26 (5.2)	NA	27 (5.0)
LTC placement at follow-up (%)	15 (3.2)	38 (7.6)	15 (2.9)	38 (7.0)
SVI (SD)	0.30 (0.13)	0.30 (0.13)	0.30 (0.13)	0.30 (0.13)
FI at baseline (SD)	0.19 (0.10)	0.19 (0.10)	0.19 (0.10)	0.19 (0.11)
FI at admission (SD)	0.25 (0.12)	0.26 (0.13)	0.25 (0.12)	0.26 (0.13)
Dementia (%)	25 (5.3)	32 (6.4)	27 (5.2)	34 (6.3)

Sensitivity analysis 1 included participants who were previously in long-term care (LTC). Sensitivity analysis 2 included all participants (e.g., transferred to other units/hospitals or died), but excluded those who were in LTC prior to admission. Sensitivity analysis 3 included all participants (e.g., transferred to other units/hospitals or died), including those who were in LTC prior to admission and controlled for previous LTC status.

**Table 3 healthcare-07-00091-t003:** Coefficients (b) and odds ratios (OR) plus 95% confidence intervals (CI) from the logistic regression results for the main and sensitivity analyses.

	Main Analyses		Sensitivity 1		Sensitivity 2		Sensitivity 3	
	b	OR	b	OR	b	OR	b	OR
SVI	0.89 (−1.22, 3.13)	2.43 (0.30, 22.86)	0.40 (−1.21, 2.35)	1.50 (0.30, 10.53)	0.81 (−1.27, 3.07)	2.26 (0.28, 21.65)	0.34 (−1.27, 2.17)	1.41 (0.28, 8.75)
FI_baseline	2.71 (0.78, 4.97)	14.98 (2.17, 144.71)	2.50 (0.79, 4.37)	12.23 (2.21, 79.05)	2.67 (0.77, 4.86)	14.50 (2.17, 129.41)	2.54 (0.81, 4.37)	12.68 (2.25, 79.42)
FI_admission	0.43 (−0.39, 1.25)	1.55 (0.68, 3.48)	0.12 (−0.67, 0.87)	1.13 (0.51, 2.40)	0.41 (−0.43, 1.23)	1.51 (0.65, 3.42)	−0.08 (−0.88, 0.67)	0.93 (0.42, 1.96)
Age	−0.24 (−0.51, 0.07)	0.79 (0.60, 1.07)	−0.26 (−0.48, 0.01)	0.77 (0.62, 1.01)	−0.25 (−0.50, 0.03)	0.78 (0.61, 1.03)	−0.26 (−0.48, −0.02)	0.77 (0.62, 0.98)
Sex (women)	2.78 (−2.19, 7.94)	16.11 (0.11, 2813.90)	2.74 (−1.19, 6.81)	15.42 (0.30, 905.87)	2.77 (−1.82, 7.68)	15.96 (0.16, 2171.94)	2.80 (−1.05, 6.84)	16.49 (0.35, 930.64)
Prior LTC	NA	NA	5.30 (3.93, 6.99)	199.61 (51.14, 1081.85)	NA	NA	5.21 (3.90, 6.80)	182.44 (49.56, 896.51)
SVI by FI_baseline	−0.70 (−1.26, −0.24)	0.50 (0.28, 0.79)	−0.65 (−1.10, −0.25)	0.52 (0.33, 0.78)	−0.69 (−1.24, −0.24)	0.50 (0.29, 0.79)	−0.64 (−1.08, −0.23)	0.53 (0.34, 0.79)
SVI by Age	0.09 (0.00, 0.16)	1.09 (1.00, 1.18)	0.10 (0.03, 0.17)	1.11 (1.03, 1.19)	0.09 (0.01, 0.16)	1.09 (1.01, 1.18)	0.10 (0.04, 0.17)	1.11 (1.04, 1.19)
SVI by Sex	−0.41 (−1.70, 1.12)	0.66 (0.18, 3.07)	−0.54 (−1.62, 0.59)	0.58 (0.20, 1.81)	−0.39 (−1.63, 1.06)	0.68 (0.20, 2.88)	−0.50 (−1.56, 0.64)	0.61 (0.21, 1.89)

Sensitivity analysis 1 included participants who were previously in LTC but controlled for previous LTC status. Sensitivity analysis 2 included all participants (e.g., transferred to other units/hospitals or died), but excluded those who were in LTC prior to admission. Sensitivity analysis 3 included all participants (e.g., transferred to other units/hospitals or died), including those who were in LTC prior to admission and controlled for previous LTC status.

**Table 4 healthcare-07-00091-t004:** Simple associations between social vulnerability and odds of LTC placement at different levels of frailty and ages for the main analyses: OR (95%CI).

Age	70 years	80 years	90 years
FI = 0.05	2.63 (0.41, 17.62)	6.20 (0.89, 43.21)	**14.64 (1.55, 127.21)**
FI = 0.15	1.30 (0.28, 5.34)	3.07 (0.62, 13.28)	**7.25 (1.06, 41.84)**
FI = 0.25	0.65 (0.17, 1.88)	1.52 (0.40, 4.46)	3.60 (0.68, 15.17)
FI = 0.35	**0.32 (0.09, 0.94)**	0.76 (0.23, 1.87)	1.78 (0.39, 6.27)

Bolded cells are statistically significant at *p* < 0.05.

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
