# Peer review of "Long-Term Care Admissions Following Hospitalization: The Role of Social Vulnerability"

_healthcare, 2019, doi:10.3390/healthcare7030091_

Round 1

Reviewer 1 Report

Can you provide the breakdown of the social frailty index ? 

Author Response

Can you provide the breakdown of the social frailty index ? 

·       Thank you for this suggestion. We have added a table that lists the 18 SVI items and coding.

Reviewer 2 Report

Dear authors,

Thank you for your work, which has been well written, with clarity and accessibility. Tables are presented with details. The key ideas of the manuscript and takeaway lessons are summarized in an intelligible manner.

I find the content of the submission compelling and useful.

Nonetheless, I would like to suggest a few points that could to a certain extent improve the usefulness and completeness of the academic discusions delivered by the paper.

1. Although at some point, the paper does mention the issue of costs, social costs, treatment costs, I still believe there should be more to discuss this critically important matter to both patients, policymakers and healthcare professionals. The problem has become increasingly acute given the changes in health policies around the world, governments' funding issues and the dominance of the private-sector health equipment and supplies providers.

Therefore, the matter of costs, such as those discussed in https://www.ncbi.nlm.nih.gov/pubmed/26413435, i.e., financial burdens and severe consequences, deserves more attention. There is evidence showing that in today's world, patients without economic privileges will have little choice but to accept costs, which is counterintuitive, and potentially detrimental to the overall level of sustainability of the system (see, https://www.nature.com/articles/s41599-018-0127-3).

2. The overall science policy in dealing with healthcare sustainability and social vulnerability should also be part of the discussion, toward the end of the paper (or even right at the beginning). The government and philanthropic activities regarding research funding and science applications will help to shape the future developments, and their influences on healthcare and social vulnerability issues will be natural. Policy failures will be a majour source of risks for the sustainability of the healthcare system (see, https://www.nature.com/articles/s41562-017-0281-4).

I believe addressing these issues will make the paper complete and even more useful for policymaking and public understanding. Thus, I hope that the authors will spend a little bit more time on this, and as a reader, I will be reading an excellent academic contribution in the near future.

Author Response

Thank you for your work, which has been well written, with clarity and accessibility. Tables are presented with details. The key ideas of the manuscript and takeaway lessons are summarized in an intelligible manner.

I find the content of the submission compelling and useful.

Nonetheless, I would like to suggest a few points that could to a certain extent improve the usefulness and completeness of the academic discusions delivered by the paper.

·       Thank you for this positive feedback of our work.

1. Although at some point, the paper does mention the issue of costs, social costs, treatment costs, I still believe there should be more to discuss this critically important matter to both patients, policymakers and healthcare professionals. The problem has become increasingly acute given the changes in health policies around the world, governments' funding issues and the dominance of the private-sector health equipment and supplies providers.

Therefore, the matter of costs, such as those discussed in https://www.ncbi.nlm.nih.gov/pubmed/26413435, i.e., financial burdens and severe consequences, deserves more attention. There is evidence showing that in today's world, patients without economic privileges will have little choice but to accept costs, which is counterintuitive, and potentially detrimental to the overall level of sustainability of the system (see, https://www.nature.com/articles/s41599-018-0127-3).

·       Thank you for drawing our attention to the importance of costs. We have added a brief discussion in the second paragraph of the introduction and have cited one of the above articles.

·       We have added an acknowledgement in the 5th paragraph of the discussion that perceptions of lengthy and costly stay in LTC could deter younger patients and family members from considering LTC as an option.

2. The overall science policy in dealing with healthcare sustainability and social vulnerability should also be part of the discussion, toward the end of the paper (or even right at the beginning). The government and philanthropic activities regarding research funding and science applications will help to shape the future developments, and their influences on healthcare and social vulnerability issues will be natural. Policy failures will be a majour source of risks for the sustainability of the healthcare system (see, https://www.nature.com/articles/s41562-017-0281-4).

·       Thank you for drawing our attention to the need to draw direct links for policy makers. We have added a paragraph to the discussion (2nd from last) that draws attention to the importance of our results to policy makers.

I believe addressing these issues will make the paper complete and even more useful for policymaking and public understanding. Thus, I hope that the authors will spend a little bit more time on this, and as a reader, I will be reading an excellent academic contribution in the near future.

Reviewer 3 Report

This is a potentially interesting paper on an issue that is not completely new but is highly relevant. I would like to have more information on:

novelty

why are data of 2012 analysed now?

are results generalizable to chronic diseases ( chronic diseases may be more affected by social issues...)?

did the authors compare the data with other influenza periods?

what can be done to reduce the effect of frailty?

what did we actually learn from this study?

Author Response

This is a potentially interesting paper on an issue that is not completely new but is highly relevant. I would like to have more information on:

Novelty

·       Thank you for alerting us that we have not clearly identified how our research is novel. Although a number of studies have examined the association between social factors and LTC, these social factors are usually considered only one at a time. The social vulnerability index (SVI) takes into consideration a broad range of social and economic circumstances into a single index. We believe that this is the first study to consider how social and economic circumstances holistically are related to LTC placement following hospitalization. Further, almost no studies have examined the moderating effects of sex, age, and frailty. We have made adjustments to the last paragraph in the introduction that clarifies how our research is original.

why are data of 2012 analysed now?

·       This is a secondary analysis of existing data from the Canadian Immunization Research Network’s Serious Outcomes Surveillance Network data. The 2011-2012 influenza season was the only season for which the data for constructing the social vulnerability index (SVI) was available. We have added a sentence to the participant section of the methods to make it clearer that we conducted secondary data analysis.

are results generalizable to chronic diseases ( chronic diseases may be more affected by social issues...)?

·       Some of our previous research has shown that social vulnerability is associated with cognitive impairment, mortality, and frailty. Although these studies demonstrate the connection between social vulnerability and a number of adverse health outcomes, they did not consider whether the effect differs by age or frailty level. Based on these previous studies, we would expect social vulnerability to be associated with some chronic diseases, but whether the association would vary based on age or frailty is unknown. This very interesting question is beyond the scope of our current paper, and we are planning to investigate this question in future research on social vulnerability.

did the authors compare the data with other influenza periods?

·       We only have data for the SVI for the 2011-2012 influenza season, therefore we are unable to compare our results with other seasons. We have added a sentence to the last paragraph of the discussion that presents this limitation.

what can be done to reduce the effect of frailty?

·       We agree that this is an important question. In our study and others, frailty is associated with LTC care. In our current study we were focused on the association of social vulnerability to odds of LTC placement. To understand how to reduce the effect of frailty is beyond the scope of the current paper and may require the completion of interventional studies. We think that this is an important avenue to explore in future research and have added a sentence at the end of our limitations section recommending that this question be address in future research.

what did we actually learn from this study?

·       We learned that the association between social vulnerability and odds of LTC placement is complex and different for different groups of people.  More specifically, the association varies depending on a person’s age and their level of frailty. Younger frail individuals who are socially vulnerable may not be able to access LTC when they need it. On the other hand, older, less frail individual who are socially vulnerable may be placed in LTC prematurely, rather than receiving the necessary supports to age in place. We have added a sentence to our concluding paragraph to make this more apparent.

Round 2

Reviewer 2 Report

Thank you for your spending time on this revised manuscript. Its readability and usefulness have improved from what I can see. I would like to congratulate you on this serious attempt.

Author Response

Thank you for your spending time on this revised manuscript. Its readability and usefulness have improved from what I can see. I would like to congratulate you on this serious attempt.

·      Thank you.

Reviewer 3 Report

The paper is clearer and more detailed, and the Authors improved it on several aspects.

I  still suggest to:

explain why an influenza episode may provide a "model of study" still of value; Why was this study undertaken at distance from the initial event?

comment on the changes in the Canadian System over these 8 years. 

be clear on what this study adds to the current literature

Author Response

The paper is clearer and more detailed, and the Authors improved it on several aspects.

I  still suggest to:

explain why an influenza episode may provide a "model of study" still of value; Why was this study undertaken at distance from the initial event?

·       We were taking advantage of existing data that was available to answer a research question we had. The availability of reliable data on LTC both prior to admission, and discharge, and 30 days post discharge in addition to the social and health variables made this an ideal dataset to answer our research question. We have added a sentence to the participants section to make this clearer.

comment on the changes in the Canadian System over these 8 years. 

·       In the context of acute care and long-term care there have been few changes to the Canadian health care system over the past 8 years. The relationship between LTC and acute care may also vary by jurisdiction within Canada. Although recently Medical Assistance in Dying (MAiD) has become legal in Canada, we do not believe this would have a substantial impact on our results. We have added a brief discussion to this point in our limitations section.

be clear on what this study adds to the current literature

·       We’re sorry that this still wasn’t clear in our revised manuscript. We have added a sentence at the end of the discussion that highlights the contribution of our study to the literature.